# Neglected Facts on *Mycobacterium Avium* Subspecies *Paratuberculosis* and Type 1 Diabetes

**DOI:** 10.3390/ijms23073657

**Published:** 2022-03-26

**Authors:** Veronika Ozana, Karel Hruska, Leonardo A. Sechi

**Affiliations:** 1Faculty of Pharmacy, Masaryk University, 612 00 Brno, Czech Republic; veronika.juzova@gmail.com; 2Orlova Department, Karvina-Raj Hospital, 734 01 Karvina, Czech Republic; 3Veterinary Research Institute, 612 00 Brno, Czech Republic; 4Institute for Research and Education, 621 00 Brno, Czech Republic; 5Dipartimento di Scienze Biomediche, Sezione di Microbiologia Sperimentale e Clinica, Università degli Studi di Sassari, 07100 Sassari, Italy; 6AOU Sassari, UC Microbiologia e Virologia, 07100 Sassari, Italy

**Keywords:** chronic inflammatory diseases, autoimmune diseases, T1D, nontuberculous mycobacteria, civilization diseases, civilization factors, global health problem, *Mycobacterium avium* subspecies *paratuberculosis*

## Abstract

Civilization factors are responsible for the increasing of human exposure to mycobacteria from environment, water, and food during the last few decades. Urbanization, lifestyle changes and new technologies in the animal and plant industry are involved in frequent contact of people with mycobacteria. Type 1 diabetes is a multifactorial polygenic disease; its origin is conditioned by the mutual interaction of genetic and other factors. The environmental factors and certain pathogenetic pathways are shared by some immune mediated chronic inflammatory and autoimmune diseases, which are associated with triggers originating mainly from *Mycobacterium avium* subspecies *paratuberculosis*, an intestinal pathogen which persists in the environment. Type 1 diabetes and some other chronic inflammatory diseases thus pose the global health problem which could be mitigated by measures aimed to decrease the human exposure to this neglected zoonotic mycobacterium.

## 1. Introduction

Type 1 diabetes (T1D) is an insulin-dependent type of diabetes caused by the destruction of pancreatic β-cells leading to major insulin deficiency. T1D represents around 10% of all cases of diabetes [1]. It develops either on an autoimmune basis, which is the cause of the disease in 70–90%, or idiopathically with not entirely clear pathogenesis [2]. The disease is most often diagnosed in children and adolescents, in whom the first symptoms are significant polyuria, polydipsia, and polyphagia. T1D may also manifest in adulthood in middle-aged and elderly patients. The disease often resembles type 2 diabetes at first, but gradually insulin production weakens, and its exogenous supply is required. The term latent autoimmune diabetes in adults (LADA) is used for this specific form. The correct diagnosis is possible to verify in the laboratory by confirming specific LADA autoantibodies and decreasing C-peptide levels [3].

The prevalence of T1D is increasing global; the incidence of T1D in children worldwide has increased at a rate from up to 5% per year since the 1970s, in particular in fast developing countries [4]. The results of the meta-analysis performed by Mobasseri et al. [5] showed that the incidence of T1D in the world is 15 per 100,000 of the population and the prevalence is 9.5 per 10 000 population. The incidence of T1D in Europe is 15 per 100,000 of the population and the prevalence is 12.2 per 10,000 people. In the Czech Republic in the last 30 years, the incidence of T1D increased more than 3 times reaching 25 new cases per 100,000 children under the age of 15. T1D is the most common diabetes type in this age (95% cases) along with monogenic diabetes (4%) and type 2 (1%) [6]. Not only T1D, but also other autoimmune diseases such as inflammatory bowel diseases, autoimmune thyroiditis and juvenile idiopathic arthritis show increasing incidence [7] without known specific triggering factors.

## 2. T1D Etiology and Environmental Factors

T1D is a multifactorial polygenic disease; its origin is conditioned by the mutual interaction of genetic and other factors. The influence of genetics on the etiology of T1D is approximately 50%, especially the strong association of T1D with genes for HLA-II (human leukocyte antigen) molecules. The role of HLA-molecules present on the surface of leukocytes is to present antigens to T-lymphocytes [8]. After a certain time, the interaction of internal and external factors can lead to the failure of immunoregulatory mechanisms. Destructive autoimmune insulitis develops, involving autoantibodies and autoreactive T-cells targeted against specific antigenic structures of pancreatic β-cells. In the first phase, the islets of Langerhans are infiltrated by leukocytes and macrophages. Subsequent activation of antigen-presenting cells (APCs: B-lymphocytes, macrophages, and dendritic cells) leads to binding of β-cell antigens to high-risk HLA-II molecules and to presenting these antigens to potentially autoreactive T-cells. Both T-lymphocytes and APCs then produce the cytokines IFN-γ, IL-1 and TNF-α, which act cytotoxically towards β-cells through the induction of apoptosis and the formation of free radicals. The clinical manifestation of diabetes occurs only after the death of 80–90% of all insulin-producing cells [8,9].

The main environmental factors triggering T1D include viral infections caused mainly by enteroviruses, early administration of cow’s milk to infants or excessive consumption of foods containing gluten and nitrates [10]. Environmental factors (e.g., diet, viruses, and chemicals) may trigger the induction of diabetes and act as primary injurious agents damaging pancreatic beta cells or stimulating an autoimmune process. Some viruses such as encephalomyocarditis virus and Mengo virus 2T may directly infect mice pancreatic beta cells. In contrast, persistent infection of cytomegalovirus and rubella virus may induce islet cell autoantibodies against a 38 kDa islet cell protein [11]. Other potential risk factors for the development of T1D are also respiratory infections in early childhood [12], enteroviruses [13] or the group B coxsackieviruses [14].

It is suggested that the gut microbiome may protect from the development of T1D by promoting intestinal homeostasis [15]. There is evidence that reduced gut microbial diversity of the *Clostridium leptum* group in children may lead to a decreased number and function of regulatory T-cells promoting the autoimmune response [16]. The lack of vitamin D supplementation in infancy increases the subsequent risk of T1D as European case-control study (The EURODIAB Sub-study 2 Study Group. 1999) has indicated [15]. Vitamin D is important in the prevention of islet cell death and might improve the survival of islet cell grafts, and it ameliorates the production of insulin. Low vitamin D levels were shown to have negative effect on β-cell function [17]. Both early and delayed introduction of gluten have been implicated in the risk of autoimmunity and T1D [10]. A study conducted by Norris et al. [18] suggests there may be a window of exposure to cereals in infancy outside which initial exposure increases autoimmune risk in susceptible children.

The TRIGR study was initiated in 2007. Extensive casein hydrolyzed formula was given to one group of children and regular commercial milk-based formula plus casein hydrolysate in a 4:1 proportion to the control group up to 6–8 months after weaning. An international double-blind randomized clinical trial of 2159 infants with human leukocyte antigen-conferred disease susceptibility and a first-degree relative with T1D recruited from May 2002 to January 2007 in 78 study centers in 15 countries [19]. The results were preliminary reported in 2011 [20] and finally in 2018. Among infants at risk for T1D, weaning to a hydrolyzed formula compared with a conventional formula did not reduce the cumulative incidence of T1D after median follow-up for 11.5 years. These findings do not support a need to revise the dietary recommendations for infants at risk for T1D [21].

Niegowska et al. [22] included 23 children at risk for T1D, formerly involved in the TRIGR study, and 22 healthy controls (HCs). Positivity to anti *Mycobacterium avium* subsp. *paratuberculosis* (MAP) peptides and homologous human peptides was detected in 48% of at-risk subjects compared to 5.85% HCs, preceding appearance of islet autoantibodies. Being MAP easily transmitted to humans with contaminated cow’s milk and detected in retail infant formulas, MAP epitopes could still be present in extensively hydrolyzed formula and act as antigens stimulating β-cell autoimmunity. Hydrolyzed milk formula does not guarantee that it is MAP free, since it can resist easily to the enzymes used and even if they was not viable, MAP components such as MAP3865 c, homologous to zinc transporter 8 (ZNT8) and proinsulin could still be present triggering the immune response.

As this review shows, many known, and as yet undiscovered biomolecules may be involved in triggering pathological pathways that lead not only to T1D but to many other chronic immune regulated inflammatory and autoimmune diseases. Causality studies could be focused on T cell response against the specific epitopes of MAP homologous to ZNT8 and proinsulin (in addition to the B cell studies) in animal models and T1D diabetes patients at risk (presence of one or more autoantibodies) and in T1D patients at onset.

*Mycobacterium* tuberculosis releases triggers just like other mycobacteria. This is evidenced by the high number of patients with comorbidity of tuberculosis and diabetes as well as other chronic inflammatory diseases [23,24,25]. For further references see Box 3 *M. avium* subsp. *paratuberculosis* in [26] and a review by Hruska and Pavlik [27].

## 3. T1D and *Mycobacterium avium* Subsp. *Paratuberculosis* (MAP)

MAP is a fastidious slow-growing mycobacterium, which infects a large range of ruminant species and is responsible for paratuberculosis (also known as Johne’s disease). It belongs to the *M. avium* complex which is a group of slow-growing mycobacteria including *M. avium*, *M. intracellulare*, or *M. chimaera* based on a gene sequence similarity. *M. avium* has four subspecies, namely *M. avium* subsp. *avium*, *M. avium* subsp. *hominissuis*, *M. avium* subsp. *paratuberculosis* and *M. avium* subsp. *silvaticum* [28]. Opportunistic infections caused by *M. avium* are among the most frequent in patients suffering from chronic respiratory infections and/or from acquired immunodeficiency syndrome [28]. However only MAP has been associated with a large number of autoimmune and inflammatory diseases and investigations all over the world are under way [29].

The target of MAP is the digestive system, infection leads to weight loss and might cause the death of the animal. This subspecies also infects animals in the wild such as red deer, rabbits, or buffalo which raises serious ecological concerns [28]. In humans, MAP has been associated with a long list of inflammatory and autoimmune diseases: Crohn’s disease, sarcoidosis, Blau syndrome, Hashimoto’s thyroiditis, autoimmune diabetes (T1D), multiple sclerosis (MS), rheumatoid arthritis, lupus, and Parkinson’s disease [26,29,30,31,32,33,34,35,36,37,38,39]. The inhalation of aerosolized MAP-contaminated manure by women in the first four weeks of pregnancy, and intrauterine transmission to the embryo, may be responsible for the development of anencephaly in the fetus. The cluster of babies with anencephaly, with a reported rate of over 60 times the national average is likely associated with application of cow’s feces to agricultural fields at the rate of 1000 gallons per acre in the rural Yakima Valley community in state Washington, USA [39].

Furthermore, MAP as a causative agent of bovine paratuberculosis, is shed in cow’s milk and has been shown to survive pasteurization. According to many epidemiologic studies, MAP is presented as a causative agent of T1D [40,41,42]. In the Sardinian and Italian population, MAP has been previously associated with T1D as an environmental agent triggering or accelerating the disease [41,42,43]. Human exposure to MAP has increased owing to the expansion of the dairy industry in developed countries as a result of dairy cattle breeding [44].

Humans are exposed to MAP also from milk and cheese from sheep and goats which suffered from paratuberculosis. For instance, Sardinia contains high numbers of MAP infected sheep and cattle, which excrete the bacteria into the environment where they persist within the protists. Environmental cycling facilitates not only the re-infection of livestock through deposition of extracted slurry from water treatment plants but also the dispersal of MAP via aerosols directly infecting human populations [45,46]. Devitalized MAP and their decay products might be therefore present in pasteurized milk and in infant formula [47,48]. The often-cited protective effect of breastfeeding tends to indicate the burden of MAP components from formula on the baby.

Municipal tap water is colonized with nontuberculous mycobacteria (NTM) which are resistant to disinfectant, heavy metal, and antibiotics [49,50]. Hence, normal water treatment processes such as filtration and chlorination prefer mycobacteria organisms by killing off their competitors. Mycobacteria grow on tap water pipes in biofilms and on plastic water bottles and survive in amoebas in soil and water [31,51,52,53,54]. Additionally, surface water and soil can expose humans to mycobacteria. Vegetables from hydroponic plants and pork from farms affected by *M. avium* pig infections may also be involved in human exposure to mycobacteria.

Comprehensive data regarding the ecology of mycobacteria and their impact on human and animal health were published by Kazda et al. [55], by Falkinham [49,52], and by Hruska and Pavlik [27].

## 4. Epitopes, Proteins and Genes of Nontuberculous Mycobacteria Associated with T1D

T1D is characterized by uplifted immune responses targeted against several autoantigens including heat shock protein 60 (Hsp60), insulin, insulinoma-associated protein-2 (IA-2), and pancreatic glutamic acid decarboxylase (GAD65) [56]. In 1994, Rabinovitch [57] noticed that destruction of pancreatic islet β-cells and insulin-dependent diabetes mellitus (IDDM) has a genetic basis with modulating effects of environmental factors. Microbial agents including certain viruses and extracts of bacteria, fungi, and mycobacteria may have a protective action against diabetes development. Protective effects of administering microbial agents, adjuvants, and a β-cell autoantigen (GAD65) may result from activation of a Th2 subset of T-cells that produce the cytokines IL-4 and IL-10 and consequently downregulate the Th1-cell-mediated autoimmune response.

Cow’s milk feeding is considered an environmental trigger of immunity to insulin in infancy. Since cow’s milk contains bovine insulin, the development of insulin-binding antibodies may occur in children fed with cow’s milk formula. This immune response to insulin may later be diverted into auto-aggressive immunity against β-cells in some individuals [58]. It has been revealed the molecular mimicry of the mycobacterial antigens with human self-epitopes which supports the theory of MAP being an infectious trigger of T1D [46,59]. According to Songini et al. [60], MAP infects the intestine and activated T-cells migrate to the pancreatic lymph nodes and to the pancreas where they attack β-cells which present antigens structurally similar to those of MAP. Thus, MAP mimicry triggers an autoimmune process. For instance, the mycobacterial heat-shock protein of MAP (Hsp65) and GAD65 expressed in the β-cells of human islets have similar amino acid sequences and conformation [32,56,60,61]. Hsp65 is a 65 kDa protein which participates in cytokine expression and stabilizes cellular proteins in response to stress or injury. Hsp65 is presented to human CD_41_ T-cells in association with multiple HLA-DR molecules [62]. Therefore, it has been proposed that MAP being the source of mycobacterial Hsp65 is an environmental trigger for T1D [32,63]. Recognition of ZnT8, proinsulin, and homologous MAP peptides in Sardinian children at risk of T1D precedes detection of classical islet antibodies. ZnT8 which is related to insulin secretion is recently identified as an autoantibody antigen in T1D. ZnT8 is a membrane protein involved in Zn^2+^ transportation expressed in insulin-containing secretory granules of β-cells and might participate in insulin biosynthesis and release, and subsequently, involved deteriorated β-cell function [64]. MAP3865c, a MAP cell membrane protein, has a relevant sequence homology with ZnT8. Furthermore, antibodies recognizing MAP3865c epitopes have been found to cross-react with ZnT8 in T1D patients [65]. Masala et al. [66] previously also reported that MAP3865c and ZnT8 homologous sequences were cross-recognized by antibodies in Sardinian T1D adults. The MAP3865c_281–287_ epitope emerged as the major C-terminal epitope recognized. Similarly, Niegowska et al. [67,68] observed increased serum-reactivity to ZnT8 transmembrane regions and their homologous MAP peptides (MAP3865c) in either Sardinian or Italian cohorts. Niegowska et al. [68] conducted the first study aimed at the evaluation of MAP being an infective agent in LADA pathogenesis. A serum-reactivity against MAP-derived peptides and their human homologs of PI and ZnT8 was analyzed in the Sardinian population. A significantly elevated positivity for MAP/proinsulin was detected among LADA patients [68].

Rosu et al. [46] confirmed the association of MAP with T1D through the detection of a mptD protein (MAP3733c) in blood plasma from T1D patients using a phage-specific sandwich ELISA method. MptD protein is a membrane protein expressed during infection stages and a significant virulent determinant since the mptD gene emerged to be an important factor for the iron uptake and metabolic adaptation of MAP required for persistence in the host [69,70]. Alongside mptD protein, Cossu et al. [69] detected a strong immune response against MAP3738c recombinant protein in T1D sera using ELISA method. It is assumed that MAP3738c protein is involved in mycolic acid biosynthesis as cyclopropanation enzyme or methyltransferase on methoxy-mycolic acids. Positive humoral immune response was revealed only in sera from T1D patients and not in T2D subjects.

Accordingly, Niegowska et al. [41,67] also proved the role of molecular mimicry through which MAP might contribute to T1D development. Since the MAP peptides identified within different proteins (MAP2404c, MAP1,4-α-glucan branching protein and MAP3865c) were characterized by sequence homology with proinsulin (PI) and ZnT8, they evaluated levels of antibodies directed against MAP epitopes and their human homologs in children from mainland Italy. Indirect ELISA to detect antibodies specific for MAP3865c/ZnT8, MAP1, 4αgbp/PI and MAP2404c/PI homologous peptide pairs was performed with positive results. Moreover, intact bacilli were isolated from certain blood samples.

A lipid-rich cell wall is involved in the virulence of mycobacteria. Moreover, the lipids exposed on the bacterial surface are highly antigenic [28]. Unlike other mycobacteria, MAP does not produce glycopeptidolipids on the surface of the cell wall but rather a lipopentapeptide (L5P) which was demonstrated to be unique for this subspecies. This L5P antigen contains a pentapeptide core, in which the N-terminal end is linked with a fatty acid [71]. Biet et al. [28] showed in their study that L5P induces a strong host humoral response involving IgM, IgG_1_, and IgG_2_ antibodies. Niegowska et al. [67] compared titers of the previously detected antibodies with serum-reactivity to L5P. It was discovered that anti-L5P antibodies appeared constantly in individuals with a stable immunity against MAP antigens. The overall coincidence in positivity to L5P and the other MAP epitopes exceeded 90%.

Other anti-MAP antibodies which were investigated using ELISA, were those against heparin-binding hemagglutinin (HBHA) and glycosyl transferase (GSD) [44,56]. Molecular characterization of the recombinant HBHA from the MAP was reported by Sechi et al. [72] HBHA plays a significant role in the adaptation of MAP to the gastrointestinal tract of ruminants. It is an adhesin important for binding of the mycobacteria to epithelial cells and other non-phagocytic cells via heparin and heparan sulfate, present on the eukaryotic cell surface [73]. GSD is an enzyme which catalyzes the glycosidic bond formation of many oligosaccharides and glycoconjugates implicated in mycobacterial cell-wall biosynthesis [74]. Sechi et al. [44] and Rani et al. [56] observed in their studies significant humoral immune responses to recombinant HBHA and GSD, and the MAP whole-cell lysate in T1D patients. However, these responses could not be indicative of an active infection since HBHA and GSD are encoded by a wider range of mycobacteria, which raised an issue of cross-reactivity with tubercle bacilli in the bacillus Calmette–Guerin (BCG) vaccinated individuals [56].

Some of the mentioned antigens/epitopes involved in T1D pathophysiology are expressed not only in MAP, but also in other nontuberculous mycobacteria that may contribute to T1D development as well. For instance, Hsp65 is also produced in *M. bovis* [75,76]. Horváth et al. [75] measured the epitope specificity of antibodies against peptide p277 of human Hsp60 and of *M. bovis* Hsp65 as well as for human Hsp60 and *M. bovis* Hsp65 proteins by ELISA. Both, anti-human- and the anti-*M. bovis* peptide p277 antibody levels were significantly higher in the diabetic children. Antibodies to two epitope regions on Hsp60 and Hsp65 were detected in high titers, the first region was similar to the sequence found in GAD65, whereas the second one overlapped with p277 epitope. A major adhesin of mycobacteria, HBHA, is encoded by a wide range of mycobacterial species. Lefrancois et al. [73] compared sequence alignment of HBHA coming from various mycobacteria. The sequences were similar in HBHA from MAP, *M. avium* subsp. *hominissuis* and *M. bovis*, while *M. smegmatis* showed slightly different structure arrangement. Similarly, GSD can be also found within different mycobacteria [74].

The control of mycobacterial infection depends on the recognition of the pathogen and the activation of both the innate and adaptive immune responses. Toll-like receptors (TLR) were shown to play a critical role in such recognition [77]. TLRs are mainly found on the surface of macrophages and dendritic cells, but they are also expressed by tissue cells in the central nervous system, the kidneys, and the liver [78]. TLRs have the ability to recognize mycobacterial antigens released by the sequestered bacilli, glycolipids PIMs, LM, LAM, lipoproteins, and other mycobacterial factors. LAMs (lipoarabinomannans) are lipoglycans ubiquitously found in the envelope of *M. chelonae*, *M. bovis* BCG, *M. avium*, *M. kansasii*, and *M. smegmatis*. LMs (lipomannans), the biosynthetic precursors of LAMs, are present in cell wall of *M. bovis*, *M. chelonae*, and *M. kansasii*. PIMs (phosphatidyl-myoinositol mannosides) are predominantly found in *M. bovis* BCG [77]. These mycobacterial components are recognized mainly by TLR2 in association with TLR1/TLR6, or by TLR4 resulting in rapid activation of cells of the innate immune system. The balance between PIM, LM, and LAM synthesis by pathogenic mycobacteria might provide pro- or anti-inflammatory, immunomodulatory signals during primary infection but also during latent infection [62,77]. In general, these signals lead to an increase in the microcellular environment concentration of proinflammatory cytokines, antimicrobial peptides, and type I IFNs. These events, involving infection and an increase in TLR activity, lead to a release of islet antigens, which are picked up by dendritic cells and presented to pathogenic T-cells. These processes are followed by β-cell destruction leading to T1D development [78,79]. According to Adamczak et al. [78], vitamin D seems to protect against T1D by reducing the TLRs’ level of activation. It was found that the expression of TLR2 and TLR4 decreases with increasing 25-hydroxycholecalciferol serum concentrations. Analogously, the expression of vitamin D receptor and vitamin D-1-hydroxylase is upregulated by activation of TLRs. Therefore, the supplementation of vitamin D might be one such potential intervention.

Recently, it has been found that single nucleotide polymorphisms in protein tyrosine phosphatase non-receptor type 2 and 22 (PTPN2/22) affect several immunity genes, leading to an overactive immune system which is involved in the pathogenic process of inflammatory autoimmune disorders. Genes for TPN2/22 are found in T-cells, β-cells, in a majority of epithelial cell types including synovial joint tissue, and intestinal tissues, where they control apoptosis and chemokine production [80,81]. These genes may thus have a fundamental role in the development of immune dysfunction and its polymorphisms are associated with rheumatoid arthritis, T1D, systemic lupus erythematosus, or Crohn’s disease [82]. A single nucleotide polymorphism in PTPN22, rs2476601, is associated with increased risk of T1D, reduced age at onset, and reduced residual β-cell function at diagnosis. It affects T-cell receptor and B-cell receptor signaling as well as other adaptive and innate immune cell processes [83]. Sharp et al. [80] demonstrated that single nucleotide polymorphisms in PTPN2/22 are found significantly in patients with Crohn’s disease and lead to an increase in T-cell proliferation due to loss of negative regulation, an increase of pro-inflammatory cytokines such as IFN-γ, and an increase of susceptibility to mycobacterial infections. MAP DNA was detected in 61% of patients with Crohn’s disease in comparison with only 8% of healthy controls. In the field of autoimmunity, the most frequently mentioned is the PTPN22-C1858T polymorphism. A correlation between PTPN22-C1858T polymorphism and increased risk of autoimmune diseases as well as bacterial infections was shown in genome-wide association studies. Li et al. [84] showed in their study that the PTPN22-C1858T polymorphism is relevant to increased susceptibility to the infection of *M. leprae*.

The BCG vaccine has recently shown a therapeutic effect for T1D. The BCG vaccine is an attenuated form of *M. bovis* originally developed 100 years ago for tuberculosis prevention. Repeated BCG vaccinations in long-term diabetics can restore blood sugars to near normal by resetting the immune system (by restoring regulatory T-cells and selectively killing pathogenic T-cells) and by increasing glucose utilization through a metabolic shift from oxidative phosphorylation, a state of minimal sugar utilization, to aerobic glycolysis, a high glucose-utilization state [85]. Kuhtreiber et al. [86] observed in their study that BCG vaccination of long-standing T1D patients, followed by a booster in 1 month, resulted in the control of blood sugar seen after a delay of 3 years. After year 3, BCG lowered hemoglobin A1c to near normal levels for the next 5 years. Dow et al. [32] proposed an alternative explanation that the positive response to BCG in T1D individuals is due to a mitigating action of BCG upon MAP that allows recovery of pancreatic function. Klein et al. also evaluated the inhibitory effect of BCG on T1D. Their results indicate that early post birth vaccination and boosting is sufficient to reduce T1D prevalence of respective cohorts. According to their work, vaccination stimulates T-regulatory cells and natural suppressor cells that inhibit the autoimmune (diabetogenic) response against β-cells [87]. 

Dow and Chan [88] recently reported that BCG has shown benefit in T1D mellitus and multiple sclerosis, autoimmune diseases that have been linked to MAP via Hsp65 and disease-specific autoantibodies. Obviously, a number of factors lend credence to the notion of a pathogenic link between environmental mycobacteria and Sjogren’s syndrome (SS), including the presence of antibodies to mycobacterial Hsp65 in SS, the homology of Hsp65 with SS autoantigens, and the beneficial effects seen with BCG vaccination against certain autoimmune diseases. Furthermore, given that BCG may protect against NTM, has immune modifying effects, and has a strong safety record of billions of doses given, BCG and/or antimycobacterial therapeutics should be studied in SS [88].

## 5. Conclusions

Mycobacteria as a source of triggers of chronic inflammatory and autoimmune diseases.

One of the possible T1D triggers is often mentioned to be MAP infection from the environment or from contaminated food. MAP molecular mimicry can induce the production of antibodies and thus affect the production of insulin. This etiology is fully consistent with the definition of T1D as an autoimmune disease. However, many molecules, epitopes, genes, and metabolic products produced or released also from other mycobacteria are associated with the development of T1D. Many of them belong to the group of the nontuberculous mycobacteria that people commonly encounter. NTM often colonize drinking water supplies and are inhaled as an aerosol mainly in showers and whirlpools. They usually do not multiply in the affected organism and do not cause pathological changes in the lungs, skin, and internal organs. However, if the long-term and massive exposure to mycobacteria encounters the host organism with a particular genetic and health predisposition, mycobacterial breakdown products may act as triggers of chronic inflammatory and autoimmune diseases, including T1D. Ongoing diseases that may have the same etiology significantly support the risk. Triggers from other pathogens may be also related to T1D as they can affect the host organism due to persistent acute infectious diseases during which these triggers may stimulate the immune system to develop chronic disease later in life.

### 5.1. Civilization Factors Involved in Human Exposure to Mycobacteria

Chronic inflammatory and autoimmune diseases are referred to as diseases of civilization. Their incidence grows in parallel with factors that are accompanied by increasing human exposure to NTM. Urbanization has significantly increased the number of people using drinking water from municipal distribution that is used for showering and hydrotherapy during which an aerosol is formed, and mycobacteria are inhaled. The severity of the consequences is associated with genetic factors, thus individuals who already have a chronic inflammatory disease or their blood relatives may be at greater risk.

Exposure to mycobacteria is also associated with a change in lifestyle. The popularity of fast food based on ground beef and pork has increased and has become part of the regular food menu. Therefore, meat contaminated with MAP or *M. avium* spp. can be for its consumers a source of mycobacterial triggers. New technologies in vegetable growing such as hydroponics in conjunction with fish farming and aeroponics may also increase consumer exposure to mycobacteria. Children who cannot be breastfed may be exposed to dead and live mycobacteria in infant formula. Additionally, the popularity of baby swimming may contribute to the mycobacterial exposure of children during the maturation of their immune systems. The number of people who frequently use indoor swimming pools has increased in general. Daily use of whirlpools, air humidifiers and water mist for cooling can be dangerous if household water is heavily colonized with mycobacteria. Jogging in cities where the air contains high concentrations of airborne dust can cause the inhalation of mycobacteria, for which solid nanoparticles are a suitable carrier. Civilization development has also influenced animal production technologies and international trade with food and animals. In a short period of several decades, paratuberculosis and the associated contamination of bovine meat and milk have spread globally. However, wild ruminants, wild pigs, camels, and buffaloes suffer from paratuberculosis as well. In some areas with a high density of large cattle farms, humans may be endangered by aerosols generated by spraying liquid manure on the field. Chronic inflammatory and autoimmune diseases affect not only countries with advanced economies, but increasingly also developing countries due to the influence of civilization factors. Nontuberculous mycobacteria are thus a global health problem.

### 5.2. Highlights for Intervention and Control of T1D

Reducing human exposure to nontuberculous mycobacteria will be a challenging and long-term task. It is crucial to thoroughly understand the triggers of immune-mediated chronic inflammatory and autoimmune diseases by physicians and to expand the study programs at universities for students of medicine, veterinary medicine, environmental studies, and food and agriculture technology. The active participation of the public is needed for successful intervention and implementation of adequate measures. Therefore, it is necessary to disseminate the knowledge patiently so that people would be able to reduce their NTM exposure effectively. For instance, breastfeeding should be preferred to formula feeding if possible. If domestic water is heavily colonized by NTM, showering should be restricted for infants. Households should have an opportunity to control the water at an affordable price.

NTM in the environment, in water, and in food are not subjected to any control. It is necessary to determine the limits on the maximum permitted number of NTM in drinking water, in the air, and in the water of indoor swimming pools, fitness centers, and hydrotherapy facilities. Operators of these facilities should disclose to their customers the results of the inspections carried out by themselves. Strict supervision by veterinary inspections at slaughterhouses, dairies, and during industrial preparation of ground beef and pork should help to identify sources of contamination while appropriate incentives should be applied to reduce the burden. Certain recommendations and orders will need to be included in technical standards and regulations to ensure occupational safety health, food safety and consumer protection. For the semi-quantitative determination of NTM, methods that allow processing of a large number of samples as well as equipment for on-site inspection (care-of-point) based on biosensors are needed [89].

Many scientists with experience in studying MAP as a cause of Crohn’s disease have long called for measures to rapidly eliminate MAP from the milk and meat supply through effective MAP control measures including biosecurity and hygiene, vaccination, and test-and-cull programs [26,29,36,37,90,91,92,93,94,95,96,97,98]. Our contribution shows on the example of T1D that interest must be aimed not only at reducing the incidence of Crohn’s disease by reducing human exposure to MAP from milk and meat, but also to many other chronic inflammatory and autoimmune diseases. Millions of people around the world suffer from chronic inflammatory and autoimmune diseases. The illnesses of these people and the economic consequences of their absence from the labor market and their costly treatment are therefore a global health and environmental problem that needs to be addressed urgently. All measures to reduce human exposure to mycobacteria should be supported by scientists, clinicians and by macro economists and gradually applied throughout the European Union and G7 countries by competent and well-informed politicians. The combat must be moved from the scientific journals and academic field to parliaments.

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
