# Peer review of "Neglected Facts on Mycobacterium Avium Subspecies Paratuberculosis and Type 1 Diabetes"

_ijms, 2022, doi:10.3390/ijms23073657_

Round 1

Reviewer 1 Report

Ozana and colleagues submitted an excellent review concerning type 1 Diabetes mellitus and Mycobacterium avium subspecies paratuberculosis.

This was an excellent review. They described a wide range of knowledge in an easy-to-understand manner. They cited more than 100 references.

However, there was no description concerning Mycobacterium avium subspecies. Mycobacterium avium has four subspecies such as avium, hominissuis, silvaticum, and paratubeculosis. Please describe why they focus on paratuberculosis.

Author Response

We thanks the reviewer for the kind words on the manuscript, description concerning Mycobacterium avium subspecies was added including the reason of the focus on paratuberculosis.

Reviewer: "However, there was no description concerning Mycobacterium avium subspecies. Mycobacterium avium has four subspecies such as avium, hominissuis, silvaticum, and paratubeculosis.

Answer: We modified the manuscript according to the suggestion given. It was added the following sentence (lines 122-130):

It belongs to the M. avium complex which is a group of slow-growing mycobacteria including M. aviumM. intracellulare, or M. chimaera based on a gene sequence similarity. M. avium has four subspecies, namely M. avium subsp. avium M. avium subsp. hominissuisM. avium subsp. paratuberculosis and M. avium subsp. silvaticum. Opportunistic infections caused by M. avium are among the most frequent in patients suffering from chronic respiratory infections and/or from acquired immunodeficiency syndrome [29]. However only MAP has been associated with a large number of autoimmune and inflammatory diseases and investigations all over the world are under way [30].

Reviewer 2 Report

Authors are some time write type 1 diabetes and some time T1D, it should be consistence. 

Line number 127: authors need to provide abbreviation for MAP.

References required (line numbers 207, 271,).

Abbreviation BCG (line number 207, author providing it in line number 315).

Authors should go through their text once again before submission as there are number of typing errors and authors don't pay attention on abbreviations. 

Please find my specific comments below.

-What is the main question addressed by the research? Is it relevant and interesting?

Authors provide a number of evidences that Mycobacterium avium subspecies paratuberculosis play a role in pathogenesis of type 1 diabetes (T1D) based on their literature search in this review. It is very interesting as this particular pathogen is present in the environment and environmental factors trigger the pathogenesis of T1D. Thus, it is very relevant and interesting topic in the field of T1D.

-How original is the topic? What does it add to the subject area compared  with other published material?

This review is very novel based on my literature knowledge.

-Is the paper well written? Is the text clear and easy to read?

Present version of this paper required some changes in language as authors have made a number of typographic errors. Also, they should cite literature when they make strong statements based on the published reports.

-Are the conclusions consistent with the evidence and arguments presented? Do they address the main question posed?

I would say yes.

Author Response

We thank the reviewer for the helpful comments which let us to improve the manuscript. In particular:

Reviewer: Authors are some time write type 1 diabetes and some time T1D, it should be consistence. 

Answer: “Type 1 diabetes” was substituted with the abbreviation T1D, thank you.

Reviewer: Line number 127: authors need to provide abbreviation for MAP.

Answer: MAP abbreviation first appeared in line 100, so it is first explained in this line.

Reviewer. References required (line numbers 207, 271,).

Answer: Requested references were added, lines 212 and 276 in the revised manuscript.

Reviewer: Abbreviation BCG (line number 207, author providing it in line number 315).

Answer: BCG abbreviation is provided in line 257. In the previous version, it did not appear first in line 207, but in line 251. Thank you.

Authors should go through their text once again before submission as there are number of typing errors and authors don't pay attention on abbreviations. 

Answer: We went through the manuscript as suggested and checked it for any mistake, modified some abbreviations and marked the changes in red.

Authors provide a number of evidences that Mycobacterium avium subspecies paratuberculosis play a role in pathogenesis of type 1 diabetes (T1D) based on their literature search in this review. It is very interesting as this particular pathogen is present in the environment and environmental factors trigger the pathogenesis of T1D. Thus, it is very relevant and interesting topic in the field of T1D.

This review is very novel based on my literature knowledge.

Answer: Thank you

-Is the paper well written? Is the text clear and easy to read?

Present version of this paper required some changes in language as authors have made a number of typographic errors. Also, they should cite literature when they make strong statements based on the published reports.

Answer:  We followed the suggestions given. We are grateful for the observations which greatly improved the manuscript.